# Cohort Study of Maternal Gestational Weight Gain, Gestational Diabetes, and Childhood Asthma

**DOI:** 10.3390/nu14235188

**Published:** 2022-12-06

**Authors:** Orianne Dumas, Anna Chen Arroyo, Mohammad Kamal Faridi, Kaitlyn James, Sarah Hsu, Camille Powe, Carlos A. Camargo

**Affiliations:** 1Université Paris-Saclay, UVSQ, Univ. Paris-Sud, Inserm, Équipe d’Épidémiologie Respiratoire Intégrative, CESP, 94807 Villejuif, France; 2Division of Pulmonary, Allergy & Critical Care Medicine, Department of Medicine, Stanford University School of Medicine, Stanford, CA 94305, USA; 3Department of Emergency Medicine, Massachusetts General Hospital, Harvard Medical School, Boston, MA 02114, USA; 4Department of Obstetrics and Gynecology, Massachusetts General Hospital, Boston, MA 02114, USA; 5Diabetes Unit, Massachusetts General Hospital, Boston, MA 02114, USA; 6Clinical and Translational Epidemiology Unit, Massachusetts General Hospital, Boston, MA 02114, USA; 7Harvard Medical School, Boston, MA 02115, USA; 8Broad Institute, Cambridge, MA 02142, USA

**Keywords:** gestational diabetes, gestational weight gain, allergy, childhood asthma, developmental origins of health and disease

## Abstract

Data on the association of maternal gestational weight gain (GWG) and gestational diabetes mellitus (GDM) with childhood asthma are limited and inconsistent. We aimed to investigate these associations in a U.S. pre-birth cohort. Analyses included 16,351 mother–child pairs enrolled in the Massachusetts General Hospital Maternal-Child Cohort (1998–2010). Data were obtained by linking electronic health records for prenatal visits/delivery to determine BMI, GWG, and GDM (National Diabetes Data Group criteria) and to determine asthma incidence and allergies (atopic dermatitis or allergic rhinitis) for children. The associations of prenatal exposures with asthma were evaluated using logistic regression adjusted for maternal characteristics. A total of 2306 children (14%) developed asthma by age 5 years. Overall, no association was found between GWG and asthma. GDM was positively associated with offspring asthma (OR 1.46, 95% CI 1.14–1.88). Associations between GDM and asthma were observed only among mothers with early pregnancy BMI between 20 and 24.9 kg/m^2^ (OR 2.31, CI 1.46–3.65, p-interaction 0.02). We report novel findings on the impact of prenatal exposures on asthma, including increased risk among mothers with GDM, particularly those with a normal BMI. These findings support the strengthening of interventions targeted toward a healthier pregnancy, which may also be helpful for childhood asthma prevention.

## 1. Introduction

Asthma is the most common chronic disease in children [1], with a prevalence of 10.5% in U.S. children [2]. Asthma is associated with a lifelong morbidity in many individuals [3] and generates a high social and economic burden [1]. Early life is increasingly emphasized as a crucial period in the development of chronic respiratory and allergic diseases, such as asthma [4]. Prenatal determinants of asthma are of particular interest as pregnancy is seen as a window of opportunity for primary prevention [4,5], especially considering maternal risk factors that can be partly modified through lifestyle and nutritional interventions, such as body mass index (BMI), gestational weight gain (GWG), and gestational diabetes mellitus (GDM) [6,7]. Although cohort studies have shown associations between a high maternal BMI before or during pregnancy and risk of childhood asthma [8,9], the independent roles of GWG and GDM remain uncertain [9,10].

Relatively few studies [11,12,13,14,15] have examined the association between GWG and childhood asthma while simultaneously investigating maternal BMI. A recent meta-analysis reported associations with both a low GWG and a high GWG and increased risk of childhood asthma or wheeze (U-shaped relationship) [9], but results partly differed in studies published since [16,17]. Although suggested associations are modest, the public health impact may be important, as a large proportion of women have inadequate weight gain during pregnancy [16,18,19]. Additional studies in large cohorts are needed to clarify the relationship between GWG, both in absolute terms and relative to recommendation (which depend on BMI), and risk of asthma in the offspring.

The prevalence of GDM has increased in the last decades in relation to a similar trend in the prevalence of maternal obesity, among other factors [20]. The association between maternal diabetes and offspring asthma have been scarcely studied, and inconsistent results have been reported [10], with either a positive association [21,22,23,24] or an absence of association [13,25,26]. Moreover, most studies were limited by the absence of distinction between maternal diabetes before or during pregnancy [21,22,23,24,25], the lack of adjustment for important confounders, such as maternal BMI [22,23,24,26], or the relatively low number of women with GDM [13,27]. In addition, although a few studies have included early pregnancy BMI as a confounder in the association between GDM and childhood asthma [13,27,28], none, to our knowledge, has examined its role as a potential effect modifier.

Finally, differences in the association between maternal BMI and asthma by sex and allergy status have been reported [13,29,30]. Asthma is a heterogeneous disease, and different asthma phenotypes (e.g., allergic and non-allergic asthma) have been shown to have different underlying mechanisms and etiologies [31]. However, these hypotheses have scarcely been examined for the association between GWG or GDM and childhood asthma [13].

In a large cohort of mothers and their children in the greater Boston (USA) area, we aimed to investigate the association of GWG and GDM with incidence of asthma in the offspring. We examined these associations according to early pregnancy maternal BMI, offspring sex, and allergy status.

## 2. Materials and Methods

### 2.1. Study Population

The Massachusetts General Hospital (MGH) Maternal-Child Cohort (MMCC) includes 48,114 mothers and their children born at MGH between 1998 and 2015, and who received care at an affiliated hospital in the child’s first years of life. The MMCC was created by linking the MGH Maternal Health Cohort and the MGH Birth Cohort. The data were obtained by linking MGH electronic health records (EHR) for prenatal visits and delivery with data from Partners HealthCare (now Mass General Brigham (MGB)) EHR for clinical data in children. The MGB Human Research Committee approved the study and waived the requirement for informed consent.

The current analysis includes mothers who received care from 1998–2010, whose children were at least 5 years old before the 4th quarter of 2015 and had at least one health care encounter (e.g., healthy child visit) between ages 3 and 5 years. For internal validity, we excluded data beginning in the 4th quarter of 2015 due to the transition from the International Classification of Diseases (ICD)-9 to ICD-10 codes in EHR; accordingly, all outcomes were defined using ICD-9 codes.

### 2.2. Asthma and Allergies

The main outcome was the incidence of asthma by age 5 years. Asthma was defined as in prior work [32] by a primary billing diagnosis (ICD-9 493.xx) on or after age 3 years but before age 5 years, or ≥2 asthma medication “events” within a 12-month period beginning on or after age 3 years but before age 5 years. “Events” are defined as coded medication entries in either inpatient or outpatient locations. Asthma medications include short-acting bronchodilators, inhaled corticosteroids, inhaled corticosteroids plus long-acting bronchodilators, and oral leukotriene modifiers. In a validation study performed on a subset of children with this asthma definition, a board-certified allergist/immunologist confirmed 92% of asthma cases on the basis of chart review [32].

Allergies were defined by the presence of atopic dermatitis or allergic rhinitis. We did not include food allergy because it cannot be accurately evaluated based on ICD-9 codes. Atopic dermatitis was defined by ≥2 billing codes for atopic dermatitis (ICD-9 691.8) or dermatitis due to food taken internally by age 3 years (ICD-9 693.1) [32]. Allergic rhinitis was defined by ≥2 billing codes for ICD-9 code 477.x. Among children with asthma, “allergic asthma” and “non-allergic asthma” were defined on the basis of the presence or absence of atopic dermatitis or allergic rhinitis. 

### 2.3. Prenatal Exposures

The primary exposures of interest were GWG and GDM. We also studied a closely related factor, maternal early pregnancy BMI, to check the consistency of the findings from the MMCC cohort with the literature. Information on maternal height and early pregnancy weight, used to evaluate early pregnancy BMI (kg/m^2^) as well as GWG and GDM, were obtained from EHR at prenatal visits and delivery.

Early pregnancy weight was evaluated, on average, at 12 weeks of gestation. Early pregnancy BMI was classified according to the following categories: <20, 20–22.4 (reference category), 22.5–24.9, 25–29.9, and ≥30 kg/m^2^ [11,13], in order to explore potential non-linear effects of maternal BMI on childhood asthma. Maternal GWG (lb) was studied first in absolute values, classified according to the following categories: <15, 15–24, 25–34 (reference category), 35–44, and ≥45 pounds gained, and then relative to early pregnancy BMI, according to the U.S. Institute of Medicine (IOM) recommendations (under recommended weight gain, meets recommended weight gain, or over recommended weight gain). The current IOM GWG guidelines (2009) recommend 28–40 lb of weight gain for underweight (<18.5 kg/m^2^), 25–35 lb for normal weight (18.5–24.9 kg/m^2^), 15–25 lb for overweight (25.0–29.9 kg/m^2^), and 11–20 lb for obese (≥30.0 kg/m^2^) pregnant women [33].

Women underwent a GDM screening test using a non-fasting 50-gram glucose loading test (GLT) between weeks 24 and 28 gestation, followed by a diagnostic fasting 100-gram 3 h oral glucose tolerance test (OGTT) if the GLT result was ≥140 mg/dl. Pregnancies were classified as having GDM if ≥2 OGTT values met or exceeded thresholds. In our analyses, GDM was defined according to (1) the Carpenter–Coustan criteria [34] and (2) the National Diabetes Data Group (NDDG, stricter) criteria. The latter was in clinical use at MGH during the study period. Pregnancies without complete GDM testing data according to the clinical protocol (e.g., missing an OGTT if GLT was abnormal, incomplete OGTT screening) were excluded from the analysis. Women known to have pre-existing diabetes were not screened for GDM and are, thus, excluded from the GDM analyses. 

### 2.4. Covariates

We considered the following maternal characteristics in multivariable models: age at delivery, history of asthma, race/ethnicity, smoking status (3 months prior to pregnancy or during pregnancy), mode of delivery (cesarean versus vaginal), and insurance status at birth (surrogate for socio-economic condition). Adjustment for self-reported maternal education was also considered, and results were similar (data not shown). Analyses of GWG and GDM were further adjusted for maternal early pregnancy BMI (kg/m^2^). As induction of labor at 38–39 weeks of gestation is indicated in women with GDM and may affect asthma risk, we adjusted for gestational age at delivery (continuous) in a sensitivity analysis.

### 2.5. Statistical Analyses

The associations of prenatal exposures with asthma in the offspring, overall and by allergy subtypes, were evaluated using logistic regression, adjusted for maternal characteristics, with generalized estimating equations (GEE) to take into account familial dependence among children of the same mother. Children with non-allergic asthma and with allergic asthma were compared with children without asthma. We performed analyses stratified by sex for all prenatal exposures and stratified by maternal early pregnancy BMI (20–24.9 vs. ≥25 kg/m^2^) for GWG and GDM, with test for interactions in multivariable models. A two-sided *p* < 0.05 was considered statistically significant. All analyses were run using SAS V9.4 (SAS Institute, Cary, NC, USA).

## 3. Results

A total of 16,772 mother–child pairs were enrolled in the MMCC cohort in 1998–2010 and received care at a MGB hospital between ages 3 years and 5 years. After exclusions due to missing values for early pregnancy BMI (*n* = 384), GWG (*n* = 30), or covariates (*n* = 7), the analytic cohort for BMI and GWG analyses included 16,351 children. GDM status could not be ascertained in 1180 pregnancies; thus, the analytic cohort for GDM analyses included 15,171 children. Participants without complete GDM screening data differed on several characteristics, including early pregnancy BMI and GWG (Appendix A).

In early pregnancy, 29% of the mothers had overweight BMI and 21% had obesity (Table 1). During pregnancy, 36% of the mothers met recommended weight gain, 40% gained more, and 24% gained less weight than recommended. GDM was observed in 710 (4.7%) pregnancies according to Carpenter–Coustan criteria and 441 (2.9%) according to the NDDG criteria. A total of 2306 children (14%) developed asthma by age 5 years, and, among them, 1525 (66%) had non-allergic asthma and 781 (34%) had allergic asthma.

In multivariable models (Table 2), early pregnancy BMI was positively associated with higher risk of offspring asthma, with an increasing trend across BMI categories (BMI 22.5–24.9: odds ratio (OR) 1.17, 95% confidence interval (CI) 1.01–1.35; BMI 25–29.9: OR 1.21, CI 1.05–1.39; BMI ≥ 30: OR 1.29, CI 1.12–1.50; *p*-trend < 0.001). A similar trend was observed for non-allergic asthma, although associations were attenuated. More pronounced associations were observed for allergic asthma, with ORs > 1.35 for all BMI categories ≥22.5. Lower early pregnancy BMI (<20) was also associated with increased risk of allergic asthma (OR 1.52, CI 1.12–2.06) compared with the reference category (BMI 20.0–22.4). Associations between maternal BMI and asthma were similar in boys and girls (Appendix A).

Overall, no association was found between GWG, in absolute values or relative to recommendations, and offspring asthma. For allergic asthma, we observed a positive association with the highest GWG category (>45 lb: OR 1.31, 95% CI 1.00–1.71; Table 2). Results were similar in boys and girls (Appendix A) and in analyses stratified according to early pregnancy BMI (Appendix A).

GDM was positively associated with offspring asthma, regardless of the definition used (Carpenter–Coustan criteria: OR 1.24, 95% CI 1.01–1.53; NDDG criteria: OR 1.46, CI 1.14–1.88; Table 3). Using the NDDG criteria, associations were more pronounced for non-allergic asthma (OR 1.57, CI 1.17–2.11) than for allergic asthma (OR 1.23, CI 0.81–1.85), although this difference was not statistically significant (*p* = 0.35). In sensitivity analyses with further adjustment for gestational age, associations were attenuated but remained significant for the NDDG criteria (Appendix A). Analyses stratified by child’s sex did not show any significant differences (p-interaction > 0.50) (Appendix A). In analyses stratified by early pregnancy BMI (Figure 1), associations between GDM and offspring asthma were observed among those with early pregnancy BMI between 20 and 24.9 kg/m^2^ (Carpenter–Coustan criteria: OR 1.64, 95% CI 1.09–2.45; NDDG criteria: OR 2.31, CI 1.46–3.65), while ORs were close to null among women with early pregnancy BMI ≥ 25 kg/m^2^ (p-interaction Carpenter–Coustan criteria: 0.09; NDDG criteria: 0.02). This difference was observed for non-allergic asthma (p-interaction < 0.05) but not for allergic asthma (p-interaction > 0.80). 

## 4. Discussion

In a cohort of >16,000 mother–child pairs, we examined the impact of several prenatal exposures, including maternal early pregnancy BMI, GWG, and GDM, on the risk of asthma in the offspring. 

Our findings confirmed a positive association between early pregnancy overweight or obesity and higher risk of asthma in the offspring, and we found an additional association between low (<20 kg/m^2^) early pregnancy BMI and allergic asthma. While an association between high maternal BMI and increased risk of asthma in the offspring is well-established, the latest meta-analysis reported an absence of association between low maternal BMI (underweight) and asthma risk [9]. However, allergic and non-allergic asthma subtypes were not distinguished. In the current study, low maternal BMI was associated only with allergic asthma in the offspring. Few studies have examined allergic phenotypes in relation to maternal BMI. In a previous U.S. study, we found no association between low maternal pre-pregnancy BMI and either allergic nor non-allergic asthma in the child [13]. A study of Danish women with asthma found that maternal underweight was associated with increased risk of wheeze in the offspring [35]. This study [35] and a few others [17,18,36] also found an association with increased risk of atopic dermatitis. It is, thus, possible that an association of low maternal BMI with atopic diseases drives the results currently observed for allergic asthma. 

Regarding GWG, the current study does not support any clear association with offspring asthma. Indeed, we found no association between GWG and asthma, except for very high weight gain (≥45 lb) and allergic asthma. Inconsistent findings have been reported in the literature on this question: while the risk of childhood asthma or wheeze has been reported to modestly increase with both low and high GWG values in a meta-analysis of six studies [9], different findings have since been reported. In a Chinese study of 15,145 mother–child pairs, a low GWG, including GWG below IOM recommendations, was associated with a lower risk of asthma [16]. In a Canadian study using linkage of birth registry and health administrative databases to form a cohort of 214,017 children, no association was found between GWG and asthma [17]. The authors from the latter study noted that a null or modest association between GWG and offspring asthma, in contrast with the well-acknowledged pre-pregnancy BMI–offspring asthma association, is consistent with recent findings of a stronger impact of pre-pregnancy BMI than GWG on inflammatory markers generally thought to influence asthma risk [17].

In the current study, we found that GDM was associated with increased risk of asthma in the offspring. Few studies have examined the association between maternal diabetes and the risk of childhood asthma. A meta-analysis published in 2021 [10], including only four studies on maternal diabetes, found a modest but significantly increased risk of asthma in the offspring (meta-RR: 1.13, 95% CI 1.01–1.27). However, most previous studies, including those in the meta-analysis, examined all types of maternal diabetes together and not specifically GDM [10,23,24,26]. Distinguishing the type of diabetes is important, as they vary in terms of etiology, severity, management, and health consequences [37]. Moreover, they have been suggested to be differentially associated with offspring wheezing/asthma [28,38]. Another limitation from previous studies was the absence of adjustment for maternal BMI, an important confounder [22,23,24,26]. A few more recent studies with stronger designs and methodology inform about a potential role of GDM in childhood asthma. In a cohort of 97,554 children aged 5 years or older in Southern California using EHR data, a 17% increased risk of asthma in children was reported for mothers with GDM requiring medication; however, the association was attenuated (HR: 1.10,96% CI: 0.99–1.21, *p* = 0.07) after adjustment for maternal pre-pregnancy BMI and GWG [28]. Similarly, a very large Danish registry-based cohort examining early-life wheezing phenotypes reported associations between GDM and early onset wheeze, which were no longer significant after adjustment for pre-pregnancy BMI [38]. In the latter study, relatively sensitive definitions for wheezing phenotypes were used (wheezing treatment at least once during 0–3 years and/or 4–6 years), which are likely to capture both early life transient wheezing and childhood asthma. In a smaller U.S. cohort (*n* = 1107, of which 62 women had GDM), GDM was associated with a significant, two-fold increased risk of asthma at age 4 years, defined by questionnaire [27]. Our findings, thus, make an important addition to the limited amount of evidence on the relationship between GDM and child asthma.

To the best of our knowledge, we are the first to examine the association between GDM and offspring asthma according to maternal pre-pregnancy BMI and child allergic status. A stronger association between GDM and offspring asthma was observed among women with normal early pregnancy BMI, in particular, for non-allergic asthma. Asthma is a heterogeneous disease, with multiple determinants and associated mechanisms. The current findings suggest a role of predominantly nonallergic pathways in the association between maternal GDM and child asthma, but this needs to be confirmed. The absence of association among women who are overweight or obese may be due to the already increased risk of offspring asthma conferred by maternal overweight or obesity, which may not be further increased by GDM. The different associations across the weight spectrum may also be related to distinct mechanisms of GDM in those with different BMI. Recent data also emphasize GDM heterogeneity [37]. Among identified GDM physiologic subtypes, only insulin-resistant GDM (approximately half of the cases) was associated with higher BMI. [37] Women with insulin-deficient GDM had BMIs similar to those of women with normal glucose tolerance. While mechanisms linking different GDM subtypes to adverse maternal and child outcomes still need to be elucidated, considering the heterogeneity of GDM when studying its causes and consequences is encouraged [37]. Interestingly, a nutritional intervention promoting a Mediterranean diet in pregnant women, first shown to reduce the risk of GDM in a randomized control trial, has been subsequently associated with a lower risk of hospitalization for asthma or bronchiolitis by age 2 years in children [7]. This finding was observed only in women with pre-pregnancy BMI <25 kg/m^2^. Whether dietary changes included in prevention and management of GDM may also be relevant for the prevention of childhood asthma merits further investigation [6].

The strengths of the current study include the large population size and the comprehensive assessment of prenatal exposures. Maternal BMI and GWG were based on objective measurements performed during prenatal visits and, thus, do not rely on mother’s recall, which may be subject to bias [39]. Maternal weight was first measured at the first prenatal visit, on average, at 12 weeks of pregnancy. Thus, an objective assessment of pre-pregnancy BMI was not available. We acknowledge that this limitation makes it more difficult to distinguish the respective role of pre-pregnancy BMI and GWG in childhood asthma. However, weight gain is generally minimal in the first weeks of pregnancy [33], and early pregnancy BMI can be considered as a good surrogate for pre-pregnancy BMI. Standardized procedures were used for the screening and diagnosis of GDM, and gold-standard laboratory data were available. Our results were robust when using different OGTT thresholds (Carpenter–Coustan and NDDG criteria), and associations were stronger when using the threshold that captures more severe hyperglycemia (NDDG criteria). With the latter threshold, in clinical use during the study period, associations remained significant after further adjustment for gestational age, controlling for a potential impact of early labor induction indicated in women with diagnosed GDM. Asthma definition was also based on medical records rather than parental report. We acknowledge that asthma is difficult to diagnose before age 5 years [31]. However, in this cohort of young children, we used a specific and validated [32] asthma definition, based on billing diagnosis and repeated prescription of asthma medication over a 12-month period between ages 3 and 5 years, in order to avoid inclusion of children with transient wheezing in the asthma group. While our results need to be confirmed with longer follow-up data, previous studies that examined age at asthma onset found that associations with maternal BMI tended to be stronger for early onset asthma [13,40], suggesting that early childhood is a relevant window to examine asthma onset in relation to prenatal exposures. We relied on ICD-9-coded billing diagnoses to define allergy status, which likely resulted in a relatively specific definition of “allergic asthma”. Our results regarding allergic status need to be confirmed in studies with more accurate data on allergic sensitization (e.g., IgE levels). Finally, information regarding some risk factors for childhood asthma, such as the exact number of siblings, was not available in this cohort. However, parity status (nulliparous yes/no) was not associated with child asthma, suggesting that the number of siblings is unlikely to be a strong confounder in this population.

## 5. Conclusions

In summary, in addition to the well-known role of maternal adiposity during pregnancy as a risk factor for asthma in the offspring, we report an additional association between low (<20 kg/m^2^) early pregnancy BMI and allergic asthma. We found no consistent association between GWG and asthma. Finally, we found that GDM was associated with increased risk of asthma in the offspring. The latter association was particularly strong among women with normal early pregnancy BMI (20–24.9 kg/m^2^). Possible differences according to allergy status were suggested in these relationships, which deserve further investigation, as they provide clues to potential underlying mechanisms. Our findings support the strengthening of interventions targeted toward a healthier pregnancy and reduction of adverse perinatal outcomes, as they may also be helpful for primary prevention of childhood asthma.

## Figures and Tables

**Figure 1 nutrients-14-05188-f001:**
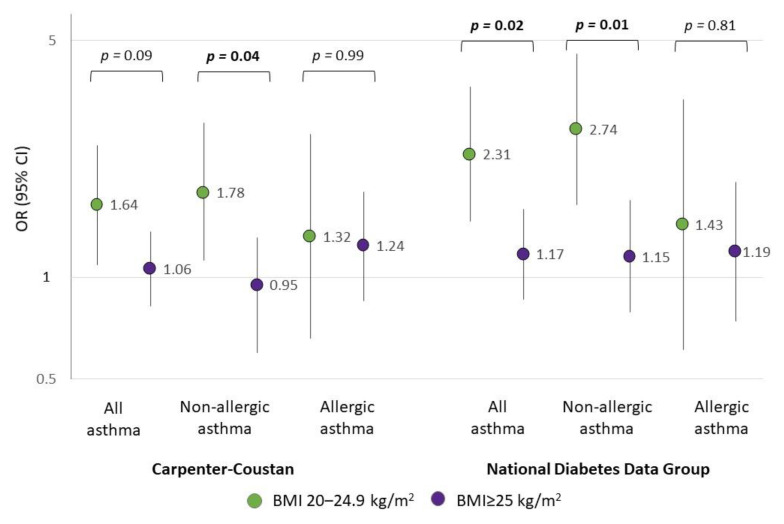
Associations of GDM with incidence of allergic and non-allergic asthma in childhood according to maternal early pregnancy BMI. BMI—body mass index; GDM—gestational diabetes mellitus; OR—odds ratio; CI—confidence interval. Analyses were adjusted for age at delivery, asthma, maternal race/ethnicity, smoking status, insurance status at birth, and mode of delivery (C-section). *p*-values for interaction are displayed on the upper part of the figure.

**Table 1 nutrients-14-05188-t001:** Maternal and child’s characteristics according to the child’s asthma status.

	All (*n* = 16,351)	Child without Asthma (*n* = 14,045)	Child with Asthma (*n* = 2306)	*p*
**Maternal characteristics**				
Early pregnancy BMI, kg/m^2^, %				<0.0001
<20.0	8.2	8.3	7.4	
20.0–22.4	19.6	20.2	16.3	
22.5–24.9	22.6	22.7	21.8	
25–29.9	28.8	28.6	30.1	
≥30	20.8	20.2	24.4	
GWG, lb, %				0.02
<15	13.8	13.5	16.0	
15–24	25.3	25.4	24.8	
25–34	34.0	34.1	33.0	
35–44	19.3	19.5	18.3	
≥45	7.6	7.5	7.9	
GWG relative to recommendations, %				0.29
Under recommended weight gain	23.8	23.8	24.3	
Meets recommended weight gain	36.4	36.6	34.9	
Over recommended weight gain	39.8	39.6	40.8	
GDM, Carpenter–Coustan criteria (missing for 1180, 7%), %	4.7	4.5	5.8	0.01
GDM, National Diabetes Data Group criteria (missing for 1180, 7%), %	2.9	2.7	4.1	0.0002
Cesarean delivery, %	28.3	27.7	32.2	<0.0001
Nulliparous, %				0.24
Yes	47.3	47.4	46.7	
No	49.6	49.6	49.7	
Missing	3.1	3.0	3.6	
Age at delivery, mean (SD)	30.3 (6.4)	30.4 (6.4)	30.0 (6.4)	0.01
Maternal race/ethnicity, %				<0.0001
White	51.2	52.1	45.8	
Black	7.6	7.4	8.9	
Hispanic	20.2	19.9	22.2	
Asian	7.6	7.6	7.3	
None of the above	13.4	13.0	15.8	
Maternal asthma, %	8.0	7.2	12.8	<0.0001
Smoking status (smoker 3 months prior to pregnancy or during pregnancy), %	7.8	7.7	8.3	0.30
Insurance status at birth, %				<0.0001
Private	55.0	56.0	49.2	
Public	34.8	33.7	41.4	
Limited	6.6	6.6	6.3	
Other	3.6	3.7	3.0	
**Child’s characteristics**				
Female, %	47.9	49.2	39.9	<0.0001
Birth weight, lb, %				<0.0001
<5.5	6.9	6.5	9.7	
5.5–6.9	26.7	26.8	25.6	
7.0–8.4	48.8	49.1	47.1	
8.5–9.9	16.1	16.2	15.6	
≥10	1.5	1.4	2.0	
Gestational age at birth, weeks, %				<0.0001
<32	1.1	0.8	2.7	
32–36	6.0	5.7	7.7	
≥37	92.9	93.5	89.6	
Atopic dermatitis, %	9.3	8.0	17.7	<0.0001
Allergic rhinitis, %	10.4	8.2	24.3	<0.0001

BMI—body mass index; GDM—gestational diabetes mellitus; GWG—gestational weight gain; % missing values is displayed for variables with >3% missing values.

**Table 2 nutrients-14-05188-t002:** Associations of maternal BMI and GWG with incidence of allergic and non-allergic asthma in childhood.

	All Asthma		Non-Allergic Asthma		Allergic Asthma	
	No. of Cases	OR (95% CI)	No. of Cases	OR (95% CI)	No. of Cases	OR (95% CI)
Maternal early pregnancy BMI, kg/m^2^						
<20.0	170	1.08 (0.89–1.32)	97	0.89 (0.69–1.13)	73	**1.52 (1.12–2.06)**
20.0–22.4 (ref.)	375	1	261	1	114	1
22.5–24.9	502	**1.17 (1.01–1.35)**	328	1.08 (0.91–1.29)	174	**1.35 (1.06–1.72)**
25–29.9	695	**1.21 (1.05–1.39)**	451	1.11 (0.94–1.31)	244	**1.44 (1.14–1.81)**
≥30	564	**1.29 (1.12–1.50)**	388	**1.24 (1.04–1.48)**	176	**1.39 (1.09–1.79)**
*p-trend*		**0.001**		**0.002**		0.16
Maternal GWG (lb) †						
<15	369	1.04 (0.90–1.21)	269	1.12 (0.94–1.32)	100	0.88 (0.68–1.13)
15–24	573	0.90 (0.84–1.06)	363	0.89 (0.77–1.02)	210	1.05 (0.87–1.28)
25–34 (ref.)	760	1	507	1	253	1
35–44	422	0.97 (0.85–1.10)	281	0.95 (0.82–1.12)	141	0.97 (0.78–1.20)
≥45	182	1.01 (0.85–1.21)	105	0.86 (0.69–1.08)	77	**1.31 (1.00–1.71)**
*p-trend*		0.84		0.20		0.14
GWG relative to recommendations †						
Under recommendations	561	1.04 (0.92–1.17)	380	1.04 (0.91–1.20)	181	1.03 (0.84–1.25)
Meets recommendations	805	1	540	1	265	1
Over recommendations	940	1.01 (0.91–1.13)	605	0.96 (0.85–1.09)	335	1.11 (0.94–1.32)

BMI—body mass index; GWG—gestational weight gain; OR—odds ratio; CI—confidence interval. Analyses were adjusted for maternal age at delivery, asthma, maternal race/ethnicity, smoking status, insurance status at birth, and mode of delivery (C-section). † Analyses were further adjusted for maternal early pregnancy BMI. Results in boldface are statistically significant.

**Table 3 nutrients-14-05188-t003:** Associations of GDM with incidence of allergic and non-allergic asthma in childhood.

	All Asthma		Non-Allergic Asthma		Allergic Asthma	
	No. of Cases	OR (95% CI)	No. of Cases	OR (95% CI)	No. of Cases	OR (95% CI)
GDM, Carpenter–Coustan criteria						
No	1980	1	1301	1	679	1
Yes	122	**1.24 (1.01–1.53)**	78	1.21 (0.93–1.56)	44	1.28 (0.93–1.76)
GDM, National Diabetes Data Group criteria						
No	2015	1	1318	1	697	1
Yes	87	**1.46 (1.14–1.88)**	61	**1.57 (1.17–2.11)**	26	1.23 (0.81–1.85)

BMI—body mass index; GDM—gestational diabetes mellitus; OR—odds ratio; CI—confidence interval. Analyses were adjusted for maternal early pregnancy BMI, age at delivery, asthma, maternal race/ethnicity, smoking status, insurance status at birth, mode of delivery (C-section). Results in boldface are statistically significant.

## Data Availability

The data that support the findings of this study are available on reasonable request from the corresponding author. The data are not publicly available due to privacy or ethical restrictions.

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
