# Peer review of "Cohort Study of Maternal Gestational Weight Gain, Gestational Diabetes, and Childhood Asthma"

_nutrients, 2022, doi:10.3390/nu14235188_

Round 1

Reviewer 1 Report

I consider that the work is very good and I must give the congratulations to authors.

Please improve the first paragraph of the discussion as it seems more apropriate for de conclusions

Please write what is the reference of the methodology for BMI in the paragraph of prenatal exposures. 

Please improve conslusions. 

Author Response

1. I consider that the work is very good and I must give the congratulations to authors.

Authors’ response: We thank the reviewer for the positive comment.

2. Please improve the first paragraph of the discussion as it seems more apropriate for de conclusions

Authors’ response: We have simplified the first paragraph of the discussion and moved the summary of the findings to the conclusion.

3. Please write what is the reference of the methodology for BMI in the paragraph of prenatal exposures.

Authors’ response: This section was clarified (l 121-125), adding a bibliographical reference and clarifying reference categories for each variable.

4. Please improve conslusions. 

Authors’ response: The conclusions were modified to include a summary of the findings.

Reviewer 2 Report

This cohort study evaluated whether gestational weight gain (GWG) or diabetes (GDM) was associated with early childhood asthma. This study emphasized the importance of prenatal determinants of childhood asthma and provided support of importance of maternal health. There are few comments and suggestions as followed:

1) Although it indicated how allergy was defined, please specify how allergic and non-allergic asthma were defined. 

2) Why was early pregnancy BMI 20-22.4 kg/m2 used for reference instead of 18.5-24.9 kg/m2 (normal weight) as the reference?

3) Table 1 should include/present characteristics of children without asthma

4) Prematurity or birth weight has been associated with childhood asthma. It should be considered in the multivariate regression analysis. 

5) Some important factors such as breastmilk-fed, history of bronchiolitis,  maternal and early use antibiotics, number of siblings, and attending of day care should be addressed or discussed. 

Author Response

This cohort study evaluated whether gestational weight gain (GWG) or diabetes (GDM) was associated with early childhood asthma. This study emphasized the importance of prenatal determinants of childhood asthma and provided support of importance of maternal health. There are few comments and suggestions as followed:

1) Although it indicated how allergy was defined, please specify how allergic and non-allergic asthma were defined. 

Authors’ response: The definition of allergic and non-allergic asthma was clarified in the method section (l. 110-111).

2) Why was early pregnancy BMI 20-22.4 kg/m2 used for reference instead of 18.5-24.9 kg/m2 (normal weight) as the reference?

Authors’ response: The category 20-22.4 kg/m2 was used as reference in order to explore potential non-linear effects of BMI, and potential associations between either low maternal BMI (<20) or modestly increased BMI (22.5-24.9) and child asthma. These points were clarified in the method section (l. 122-123).

3) Table 1 should include/present characteristics of children without asthma

Authors’ response: Characteristics of children without asthma are presented in the 3rd column of Table 1. We have clarified the Table’s column title to make it more apparent.

4) Prematurity or birth weight has been associated with childhood asthma. It should be considered in the multivariate regression analysis. 

Authors’ response: Prematurity was taken into account in our analyses, as we adjusted for gestational age (continuous, as clarified l. 150) in a sensitivity analysis (table E4). This analysis was performed as induction of labor at 38-39 weeks of gestation is indicated in women with GDM and may affect asthma risk, and thus it was considered as a potential confounder.

On the other hand, we consider that birth weight might be a mediator rather than a confounder in the association of interest (besides the impact of prematurity, which is accounted for in sensitivity analyses), and thus it should not be adjusted for.

5) Some important factors such as breastmilk-fed, history of bronchiolitis,  maternal and early use antibiotics, number of siblings, and attending of day care should be addressed or discussed. 

Authors’ response: Please find below our response relative to each of the factors mentioned by the reviewer:

Number of siblings: The exact number of siblings was unknown in this population, which we acknowledge is a limitation. However, parity status (nulliparous yes/no) was not associated with child asthma, suggesting that the number of siblings is unlikely to be a strong confounder in this population. This point was added to the discussion, l. 339-342.

History of bronchiolitis: we consider that history of bronchiolitis might be a potential mediator rather than a confounder, and thus should not be adjusted for.

Breastmilk-fed, use antibiotics, and attending day care: although these factors may impact asthma in childhood, they cannot directly impact prenatal exposures. They might be associated with prenatal exposures due to confounding by SES/education level but the latter factors were adjusted for in the analyses.

Round 2

Reviewer 2 Report

The authors have answered all the questions. I do not have further suggestions. Thank you!